

# Element/Ca ratios in Nodosariida (Foraminifera) and their potential application for paleoenvironmental reconstructions

Laura Pacho[1]; Lennart de Nooijer[1]; Gert-Jan Reichart[1,2]

[1] Royal Netherlands Institute for Sea Research (NIOZ) and Utrecht University, Texel.
[2] Department of Geosciences, Utrecht University, Utrecht.

*Correspondence to*: Laura Pacho Sampedro (laura.pacho.sampedro@nioz.nl), Lennart de Nooijer (lennart.de.nooijer@nioz.nl), Gert-Jan Reichart (gert-jan.reichart@nioz.nl).

**Abstract.** The chemical composition of foraminiferal shells is a well-known tool in paleoceanography to reconstruct past

environments and climate. Their application is based on the relation between environmental variables and the concentration of elements incorporated or stable isotope fractionation during calcification. The vast majority of these so-called proxy-relationships are based on the foraminiferal order of the Rotaliida, that for example, encompass all living planktonic species. However, there are more orders of foraminifera with calcifying members, some of which that have fundamentally different biomineralization pathways, such as the Nodosariida, the Polymorphinida and the Vaginulinida. All these belong to the class

of the Nodosariata and produce calcite shells, which may serve as carriers of paleo-environmental and climate signals. The microstructures of these shells and overall morphology of these foraminifera strongly deviate from the Rotaliida, suggesting that their elemental and stable isotopic composition do not necessarily respond similarly to environmental parameters. A potential advantage of the Nodosariata is that they appear considerably earlier in the fossil record (Carboniferous) than the Rotaliida (Jurassic), thereby possibly extending the range of foraminifer-based paleoceanographic reconstructions

considerably. To test the potential application of Nodosariata foraminifera as paleoproxies, we investigated incorporation of 5 elements in 11 species as a function of environmental parameters from a transect sampled in the Gulf of Mexico. Their element composition (B/Ca, Na/Ca, Mg/Ca, Sr/Ca and Ba/Ca) shows a distinct geochemical signature for these foraminifera different to that of members of other foraminiferal orders. Results also show an increase in Mg/Ca values with increasing temperature, similar to that know for the Rotaliida, which suggest that Nodosariata shells might be useful for

paleotemperature reconstructions. The difference in Mg/Ca-temperature calibration in Nodosariata compared to Rotaliida, with the large differences in their morphology, shell's microstructures and overall geochemical composition, suggests that



the Mg/Ca to temperature relationship is partly independent of the exact calcification mechanism. We compare Mg/Ca-temperature sensitivities across foraminiferal orders and describe a relationship between the average Mg/Ca and the sensitivity of the Mg/Ca-temperature calibration. For other elements, the variability across orders is smaller compared to that

in Mg/Ca, which results in more similar El/Ca-environmental calibrations.

## 1. INTRODUCTION

Reconstructing past climates is an integral part of predicting the impact of the ongoing rise in atmospheric $CO_2$ levels on the Earth's future climate. The expected temperature increase for a doubling in $p$$CO_2$ (the so-called climate sensitivity) has, for instance, been estimated by comparing past seawater temperatures and carbon dioxide levels (Rosenthal et al. 2017;

D'Arrigo, Wilson, and Jacoby 2006; Mann, Bradley, and Hughes 1998). Reconstructions of such parameters rely on accurate and precise tools that can be applied to past episodes in Earth's history with conditions similar to those expected in the future. In this context, foraminifera are popular tools as they are proxy signal carriers for constraining past seawater temperature and pH. Field and culturing studies have shown the dependence of the chemical composition of their shells on the seawater chemistry and physics in which they calcified. For example, the amount of incorporated Mg (expressed as the

shell's Mg/Ca) increases exponentially with temperature (Nürnberg, Bijma, and Hemleben 1996) and can hence be used to reconstruct past changes in bottom water temperature using benthic foraminifera (Lear, Rosenthal, and Slowey 2002) and sea surface temperature using planktonic foraminifera (Hastings, Russell, and Emerson 1998; D. W. Lea, Pak, and Spero 2000). The incorporation of Mg into the calcite of most foraminifera is, however, much lower compared to calcite precipitated inorganically from seawater (Morse, Arvidson, and Lüttge 2007). This offset and the observed differences in Mg/Ca values

between species (Wit et al. 2012) is hypothesized to be caused by the strong biological control that foraminifera exert on the chemistry of the calcifying fluid from which they form their shells (Erez 2003; L J De Nooijer et al. 2014). This biological control and the resulting inter-species variability in calcite chemistry has highlighted the need for species-specific calibrations (Wit et al. 2017; Allen et al. 2016). With this in mind, Mg/Ca and other proxies based on foraminiferal shell composition, including Na/Ca for salinity (Dämmer et al. 2020; Bertlich et al. 2018; Wit et al. 2013) and $\delta^{11}$B for seawater

pH (Foster and Rae 2016; Rae et al. 2011; Spivack, You, and Smith 1993) have been developed and successfully applied. Another complicating factor when applying foraminiferal proxy signals is the dependency of element incorporation and

isotope fractionation on more than one environmental parameter. For example, shell Mg/Ca values are also affected by the marine inorganic carbon system (Evans et al. 2016), salinity (M. Raitzsch et al. 2010; Dissard et al. 2010) and the [$Mg^{2+}$] of the seawater (Evans and Mller 2012). Ideally, proxy application would therefore include multiple elements to reconstruct a

single parameter. Or alternatively, use multiple proxy relationships to simultaneously reconstruct multiple environmental parameters.

Our knowledge of the controls on foraminiferal shell geochemistry is almost exclusively based on results obtained from Rotaliida. These foraminifera are characterized by multilocular shells composed of bilamellar calcite (Reiss 1957; 1963) than can be optically radial or granular. The popularity of using members of this order is partly due to the fact that they

encompass all extant planktonic foraminiferal species, while the diversity and overall high abundance of benthic Rotaliida add to their popularity for reconstructions of bottom water conditions. Few studies investigated element incorporation in the Miliolida, which have a fundamentally different calcification mechanism (ter Kuile, Erez, and Padan 1989; Lennart Jan De Nooijer, Toyofuku, and Kitazato 2009; Debenay, J.-P., Guillou, J.-J., Geslin, E., Lesourd, M. and F. 1998). The composition of their calcite is markedly different from that of the Rotaliida, with for example markedly high Mg/Ca (Toyofuku et al.

2000b) and more depleted $\delta^{25}$Mg values (Dämmer et al. 2021) (Toyofuku et al. 2000b; I. van Dijk et al. 2017).

Reconstructions based on Rotaliida could theoretically span the last ~190ma since they first occur in the fossil record in the Pliensbachian (Haynes 1981b). The order of Nodosariata evolved calcification much earlier in the Permian (Haynes 1981a) and their application would therefore roughly double the age for which paleoceanographic reconstructions could be made using foraminiferal shell chemistry. They separated from the Rotaliida and Miliolida likely before the Cambrian (Pawlowski

et al. 2003) and are currently found in many marine habitats and are easily (Haynes 1981a) recognizable by their uniserial chamber arrangement (Haynes 1981a). Their walls are fibrous, composed of conical bundles of one to tens of μm in length, specific to the Nodosariata and therefore suggests a unique biomineralization mechanism (Dubicka, Owocki, and Gloc 2018).

So far, the Nodosariata elemental and isotopic composition has not been studied and therefore, we analyzed element

incorporation such as Na/Ca, Mg/Ca, Sr/Ca, B/Ca and Ba/Ca of different species collected along depth transect in the Gulf of

Mexico. Accompanying environmental data (temperature, salinity, etc.) allow us to detect any dependencies of the incorporation of elements on these parameters and compare them to existing calibrations for Rotaliida foraminiferal species.

## 2. MATERIAL AND METHODS

### 2.1 Environmental setting

In February 2020, sediment samples were collected from the continental margin in the northern Gulf of Mexico using research vessel *Pelagia* (expedition 64PE467). Samples were collected along a transect close to the outflow of the Atchafalaya River, at depths of 105m, 272m and 619m (Fig. 1). From box cores, smaller sub-cores were collected on deck and subsequently sliced (with a resolution of 0.5 cm for the upper two centimeters and in 1 cm-slices down to a depth of 10 cm). The sediment was stored in ethanol, with Rose Bengal (rB) (2 g/L) added to stain the cytoplasm of living foraminifera.

From the overlying water of the box cores, vials were filled for analysis of DIC (dissolved inorganic carbon) and TA (total alkalinity) after filtering over 0.4 µm filters. For both analyses, 5 mL vials were filled with seawater and stored at 4 °C after addition of 15 µL of $HgCl_2$ to prevent biological alteration of the inorganic carbon system. The samples were analyzed after returning to the laboratory using a QuAAtro Continuous Flow Analyser. DIC samples were acidified and the carbon dioxide that was formed was dialyzed over a membrane that reduces the phenolphthalein indicator and was spectrophotometrically

recorded at 550nm (Stoll et al. 2001). For TA, a slightly acid buffered solution of Potassium Hydrogen Phthalate was added to the sample after which the intensity at 590nm was recorded (Sarazin, Michard, and Prevot 1999). Values obtained for DIC and TA were consistent with the ones obtained from earlier expeditions (Sirois 2017). Bottom water temperature and salinity were taken from CTD casts at the same station where the sediment samples were taken approximately 4 meters above the sea floor. Values for all inorganic carbon system parameters can be estimated using two measured parameters, since any

combination of two such parameters will allow calculating all others, including dissolved $CO_2$ (Zeebe, Bijma, and Wolf-Gladrow 1999), performed with PyCO$_2$SYS (Lewis, E; Wallace, D; Allison 1998), using the recent published Python script (Humphreys et al. 2022).





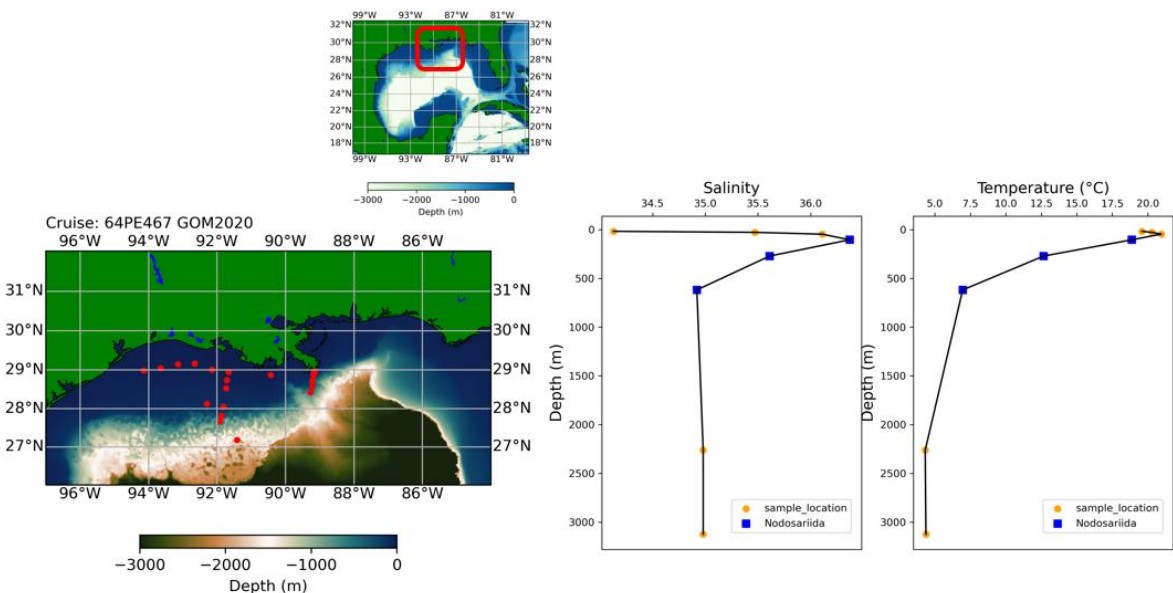

Figure 1: Left: sampling location in the Gulf of Mexico ( **G. B. C. G.**, 2022). Right: salinity and temperature data from CTD at bottom sample.

| Position | Station | Depth (m) | Temperature (°C) | Salinity | DIC (µmol/Kg) | TA (µmol/Kg) | $p\text{CO}_2$ (ppm) | pH | $[\text{CO}_3^{2-}]$ (µmol/Kg) | $[\text{HCO}_3^-]$ (µmol/Kg) | $f\text{CO}_2$ (ppm) | $\Omega\text{Ca}$ |
|---|---|---|---|---|---|---|---|---|---|---|---|---|
| 91.812W/ 28.052N | A100 | 104.7 | 18.87 | 36.37 | 2151.7 | 2334.7 | 621.60 | 7.88 | 135.84 | 1995.29 | 619.46 | 3.16 |
| 91.862W/ 27.812N | A300 | 271.64 | 12.66 | 35.61 | 2144.9 | 2277 | 623.25 | 7.86 | 102.47 | 2017.51 | 620.93 | 2.32 |
| 91.92W/ 27.665N | A600 | 618.8 | 6.96 | 34.92 | 2202.1 | 2300.7 | 616.54 | 7.84 | 82.51 | 2089.71 | 614.07 | 1.75 |

Table 1: Chemical and physical seawater parameters at the stations where foraminifera were collected. Temperature, salinity, DIC and TA were measured; the other parameters ($p\text{CO}_2$ and all parameters to the right of $p\text{CO}_2$) were calculated.

Samples were washed using sieves with mesh sizes of 63 µm and 150 µm and dried in an oven at 60°C. When selecting the specimens, rB-stained foraminifera were separated from non-stained specimens to allow detection of possible post-mortem alteration of the primary geochemical signal. The foraminifera were cleaned after isolation from the sediment by immersion
in a solution of 1% $H_2O_2$ and 0.1M $NH_4OH$, and three consecutive rinses with double deionized water. During the latter step, the Eppendorf tubes were placed in an ultrasonic bath to remove any particles adhering to the shells. In this way, 188 individuals were prepared for single-chamber geochemical analysis using laser ablation-inductively coupled plasma-mass

spectrometry (LA-ICP-MS).

Specimens were ablated for 50 seconds in a NWR193UC dual volume chamber using circular spots of 80 µm set at a repetition rate of 6 Hz, using an energy density of 1.00 ± 0.05 $J/cm^2$. The aerosol produced during the ablation was transported to a quadrupole ICP-MS (Thermo Fisher Scientific iCAP-Q) on a helium flow with a flow rate of 0.6 L/min, with 0.4 L/min Argon make-up gas being added before entering the ICP torch. Calibration was performed against USGS MACS-3

(synthetic calcium carbonate) pressed powder standard with $^{43}Ca$ as an internal standard. Scanned masses include $^{11}B$, $^{23}Na$, $^{25}Mg$, $^{27}Al$, $^{88}Sr$ and $^{138}Ba$. Repeatability based on RSDs of measurements of NFHS-2-NP (n=8) is 4% for Na/Ca, 1% for Mg/Ca and 2% for Sr/Ca (Boer et al. 2022). For all specimens, only the final chamber was ablated (Fig. 2).

Since the foraminifera were either alive or recently living when sampled, a reductive cleaning step was not applied, as oxidative coatings are assumed to form post mortem. To scan for possible contamination by clay particles, we tested for a

possible correlation between Mg/Ca and Al/Ca count ratios. Since they were not correlated and the occasional high Mg/Ca was not accompanied by high Al/Ca, the elevated Mg/Ca could not be attributed to contamination (e.g. by clay particles or a recrystallized phase at the surface of the shells) we did not remove any of the original data points. Instead, we statistically tested for outliers to identify El/Ca ratios that are outside the expected distribution given the data. These outliers (n =25) are highlighted in the figures and in- or excluding them was found to have an insignificant effect on the reported regressions.





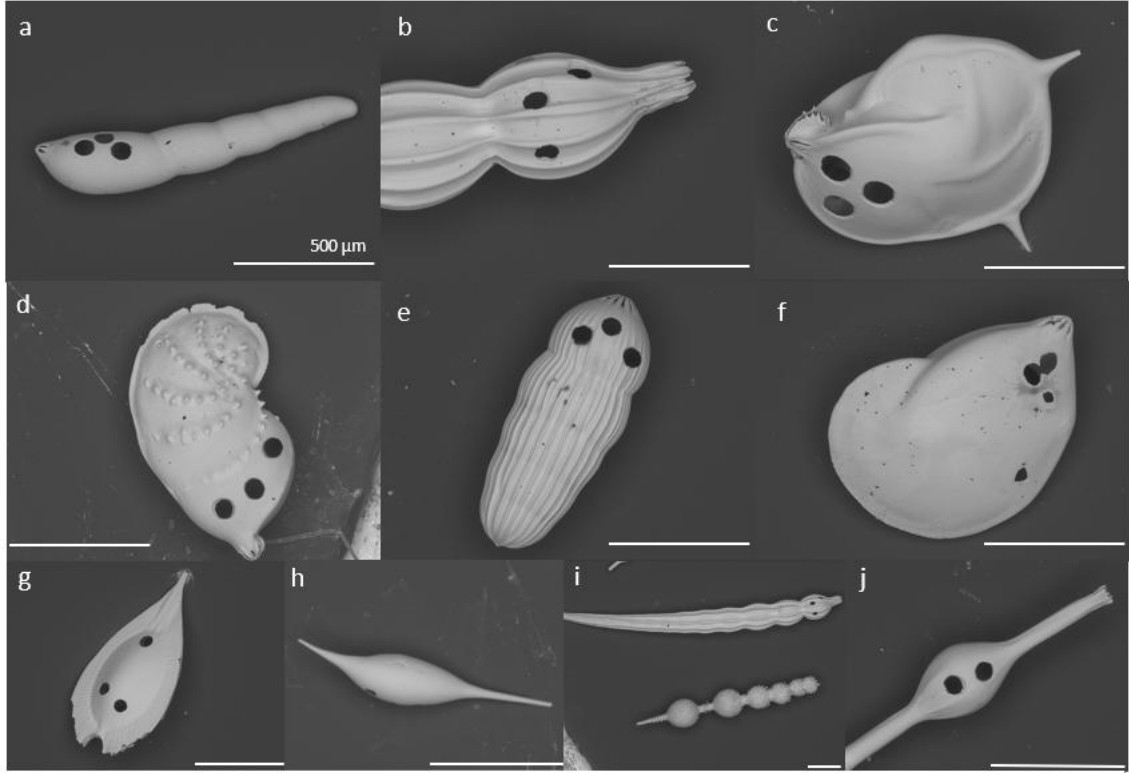


**Figure 2: Nodosariata species studied. (a):** *Dentalina* **spp.; (b)** *Nodosaria flintii*; **(c)** *Lenticulina calcar*; **(d)** *Lenticulina papillosa*; **(e)** *Pseudoglandulina comatula*; **(f)** *Lenticulina calcar*; **(g)** *Fissurina radiata*; **(h)** *Procerolagena* **sp.(i)** *Nodosaria flintii* **up and** *Amphicoryna* **sp. bottom; (j)** *Grigelis semirugosus*. **(The SEM images were taken after LA-ICPMS, holes in the pictures are a consequence of the analyses). All scale bars are 500 μm, the size of the ablation craters are all 80 μm.**

**2.2 Statistical analyses**

For the three species present at all depths sampled we performed an ordinary least sum of squares regression analysis to test dependency of the elements incorporated on environmental parameters. For Mg/Ca and temperature, an exponential response model was assumed, while assuming a linear response model for Na/Ca and salinity. For B/Ca and the inorganic carbon parameters, a linear response model was assumed. Prior to regression analysis, outliers were identified based on studentized

residuals using package "statsmodels" for python and applying the method "sidak" from the Holm-Šídák method with one-step correction (Seabold and Perktold 2010) Identified outliers are highlighted in the figures: their presence or absence had only a marginal effect on the regression analysis.





When plotting the results, the analytical error was plotted for individual analyses (which mostly falls within the size of the sample marker), the standard deviation for the sample to identify the variability within the sample as well as the standard

error (SE), to show confidence interval for the estimate of the average. Standard deviation (SD) and SE are related according

to $SE = SD/\sqrt{(n^{\circ}\ measurements)}$).

Since all data are derived from three locations, in addition to the regression analysis, a two-tailed t-test, was performed to test whether the variances of the El/Ca between locations significantly differed.

## 3. RESULTS

**3.1 Average El/Ca in Nodosariata species**

Combining the data from all stations shows that the average El/Ca in the individual Nodosariata species varies between 6.65 to 13.2 mmol/mol for Na/Ca, between 5.94 and 20.1 mmol/mol for Mg/Ca and between 1.09 and 1.81 mmol/mol for Sr/Ca (Table 2). For all measured Mg/Ca of a single species, the SD is on average 3.01 mmol/mol, where it is 1.07 mmol/mol for Na/Ca, and 0.14 mmol/mol for Sr/Ca. This translates to a relative variability in El/Ca within a species of 19.6% for Sr/Ca,

% for Na/Ca and 56% for Mg/Ca.

For the Nodosariata Ba/Ca varies considerably, between 2.5 and 4.6 µmol/mol, with an average SD of 1.2 µmol/mol, which corresponds to a 71% variability. The B/Ca data varies between 51 and 83 µmol/mol, with a modest variability in the Nodosariata, with an average SD of 9.8 (or 31.2%) per species.

| Species | Na/Ca (mmol/mol) | SD | Mg/Ca (mmol/mol) | SD | Sr/Ca (mmol/mol) | SD | Ba/Ca (µmol/mol) | SD | B/Ca (µmol/mol) | SD | Number of measurements/specimens |
|---|---|---|---|---|---|---|---|---|---|---|---|
| *Amphycorina* sp. | 8.50 | ± 0.74 | 5.94 | ± 0.79 | 1.20 | ± 0.07 | 3.18 | ± 0.74 | 53.37 | ± 5.67 | 52/19 |
| *Nodosaria flintii* | 8.22 | ± 0.85 | 8.08 | ± 2.91 | 1.30 | ± 0.14 | 3.78 | ± 1.85 | 53.09 | ± 10.13 | 30/10 |
| *Dentalina* spp. | 9.28 | ± 1.63 | 11.74 | ± 6.49 | 1.33 | ± 0.19 | 4.63 | ± 2.88 | 66.73 | ± 17.84 | 107/35 |
| *Fissurina radiata* | 6.65 | ± 0.41 | 7.42 | ± 2.92 | 1.09 | ± 0.07 | 2.53 | ± 0.25 | 53.98 | ± 6.64 | 5/2 |
| *Grigelis semirugosus* | 10.85 | ± 1.50 | 11.78 | ± 4.04 | 1.57 | ± 0.13 | 2.85 | ± 0.79 | 62.21 | ± 6.20 | 32/12 |





| | | | | | | | | | | |
|---|---|---|---|---|---|---|---|---|---|---|
| *Lenticulina calcar* | 10.71 | ± 1.19 | 10.87 | ± 2.43 | 1.58 | ± 0.15 | 2.50 | ± 1.10 | 82.72 | ± 15.06 | 104/33 |
| *Lenticulina denticulifera* | 10.03 | ± 1.21 | 11.31 | ± 2.82 | 1.54 | ± 0.18 | 3.45 | ± 1.40 | 67.29 | ± 16.87 | 85/27 |
| *Lenticulina papillosa* | 9.50 | ± 0.81 | 8.81 | ± 1.88 | 1.54 | ± 0.08 | 2.73 | ± 0.64 | 51.12 | ± 6.54 | 58/20 |
| *Procerolagena gracillini* | 10.47 | ± 1.35 | 8.64 | ± 0.93 | 1.81 | ± 0.25 | 2.85 | ± 0.76 | 64.36 | ± 3.84 | 6/2 |
| *Pseudoglandulina comatula* | 13.21 | ± 0.93 | 20.06 | ± 3.19 | 1.60 | ± 0.07 | 4.28 | ± 1.58 | 80.04 | ± 9.38 | 38/13 |
| *Oolina* spp. | 10.01 | ± 1.20 | 12.14 | ± 4.75 | 1.39 | ± 0.24 | 2.88 | ± 0.66 | 58.12 | ± 9.96 | 8/2 |

**Table 2: Element incorporation average and SD for eleven species spanning 188 analyzed specimens of Nodosariata.**

## 3.2  Impact of temperature on Mg/Ca and salinity on Na/Ca

Mg/Ca correlates positively with temperature in *Nodosaria flintii*, *Dentalina* spp*.* and *Lenticulina calcar* over a the 12 °C

range studied here. Lowest and highest values for Mg/Ca were found in *Dentalina spp.*, ranging from 3.09 mmol/mol at

600m depth to 29.2 mmol/mol at 100m depth (Fig. 3b and Table 2). For *Nodosaria flintii*, Mg/Ca ranges from 3.93

mmol/mol to 14.9 mmol/mol (Fig. 3c and Table 2) and for *Lenticulina calcar* Mg/Ca increases from 6.6 to 18 mmol/mol (Fig.

3 and Table 2). Despite the differences in absolute values, the sensitivity of changes in Mg/Ca as a function of temperature

is similar for the three species, with *Dentalina* spp. having a slightly higher Mg/Ca-temperature sensitivity than the other

two species (Table 2).



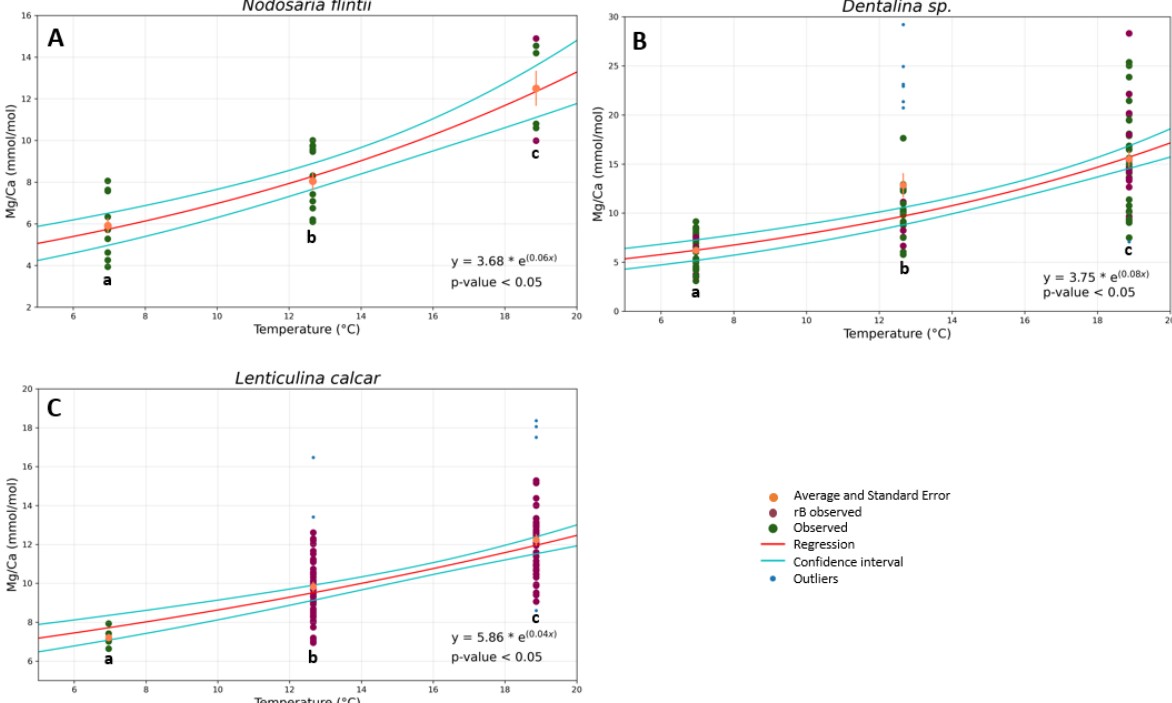

**Figure 3. Correlations between Mg/Ca and temperature for three species of Nodosariata. Dots in blue show the outliers that were omitted before regression analysis based on studentized residuals. Red points are individuals living when sampled (i.e. stained with rose-Bengal) and green points are non living individuals when sampled. Red line indicates the result of the exponential OLS regression using both stained and non stained specimens, cyan lines indicate the 95% confidence interval of the regression.**

Results also show a significant positive increase in Na/Ca with salinity for two species, *Dentalina* spp. and *Nodosaria flintii*,

despite the relatively small range in salinities between sites (1.45 units). For *N. flintii*, average Na/Ca varied between 6.7

and 10 mmol/mol, which is similar to the increase of 5.6 to 13 mmol/mol by *Nodosaria flintii* (Fig.4).



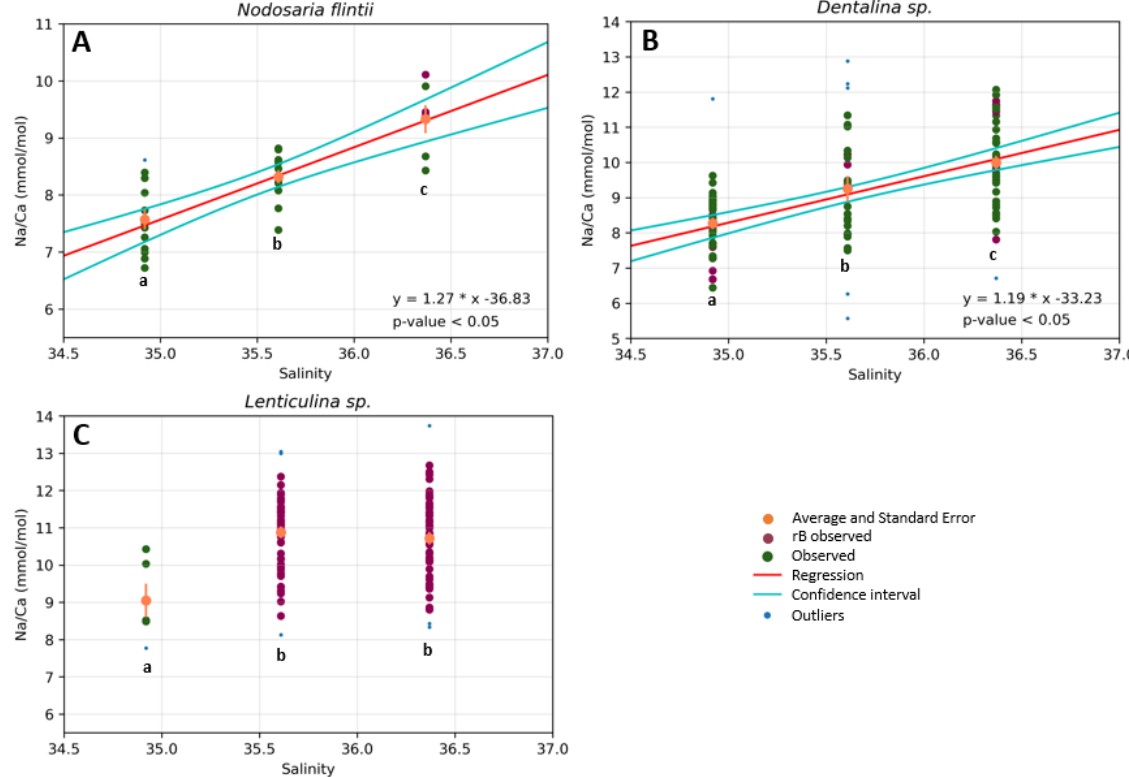

**Figure 4: Correlations between Na/Ca and salinity for three species of Nodosariata. Outliers are shown in blue that were omitted before regression analysis (panels A and B) based on studentized residuals. Red points are individuals living when sampled (stained with rose-Bengal) and data in green are non stained individuals. Red lines in A and B indicate the result of the linear OLS regression using both the stained and non stained specimens and the two cyan lines indicate the 95% confidence interval of the regression.**

For *Lenticulina* sp., no regression was found between salinity and Na/Ca (panel C); 'a' and 'b' indicate significant differences in the averages between the three groups of datapoints (i.e. the average Na/Ca at lowest salinity is significantly different from the average Na/Ca in the other two sampled stations).

## 4. DISCUSSION

### 4.1 Element incorporation in Nodosariata shells

The average values of Na/Ca, Sr/Ca and Ba/Ca of the Nodosariata are similar to those observed in planktonic (Barker et al. 2005) and many benthic Rotaliida (Lear, Rosenthal, and Slowey 2002; Elderfield et al. 2006), whereas Miliolida have

considerably lower Na/Ca and higher Sr/Ca and Ba/Ca (Inge Van Dijk, Nooijer De, and Reichart 2017). In addition, the Nodosariata's Mg/Ca, is higher (5-30 mmol/mol; Fig. 3) than those in planktonic and many low Mg/Ca benthic Rotaliida (Allen et al. 2016; Barker et al. 2005; Lowenstein and Hönisch 2012). Many larger benthic, tropical Rotaliida and most Miliolida, have an order of magnitude higher Mg/Ca values(Evans, Müller, and Erez 2018; Toyofuku et al. 2000a) compared to those observed for the Nodosariata. The Nodosariata's B/Ca values are considerably lower compared to those reported for

other foraminiferal species (Rae et al. 2011). A similar low B/Ca value was found for the rotaliid *Operculina umbonatus* (Rae et al. 2011), but this species has much higher Mg/Ca compared to the species measured here (Fig. 3). Together, our measurements imply that the Nodosariata have a unique El/Ca signature that can easily be distinguished from that of the other calcifying foraminiferal orders (Rotaliida, Miliolida, etc.), despite overlap of some ratios with those of some other species.

This order or class specific signature of the shell's composition supports a fundamental differences in their calcification mechanisms (Dubicka, Owocki, and Gloc 2018; Dubicka and Gorzelak 2017). Such a difference is also suggested with a fundamentally distinct morphology (i.e. chamber arrangement), as well as the μm scale structures observed within the chamber walls. The chamber walls of the Nodosariata show a lamellar and fibrous texture, while Rotaliida show a granular texture (Dubicka, Owocki, and Gloc 2018). Such differences in the shell's microstructure coincide with the here observed

contrasting Mg/Ca values, similar to what was already reported for comparisons between other pairs of foraminiferal orders (I van Dijk et al. 2016; Bentov and Erez 2006). Deep evolutionary branching between Nodosariata and Rotaliida and the large difference in first fossil occurrence further supports the hypothesis that they evolved their biomineralization mechanism independently. With different seawater chemical conditions(I van Dijk et al. 2016; Tanner et al. 2020) at the time when Nodosariata and Rotaliida evolved calcification biomineralization mechanisms may well reflect contrasting selective

pressures, which in turn is reflected by the shells' Mg/Ca ratios.

Although they have a long geological and evolutionary history, variability within the Nodosariata is remarkably small compared to that observed in e.g. the Rotaliida. The Mg/Ca values vary between the different families -the Lagenidae,

Nodosariidae, Ellipsolagenidae and the Vaginulinidae- not significantly (one-way ANOVA, p-value > 0.05 and F=1.082).

Also, for the other elements no significant difference in the average elemental ratios is observed between species and thus,

between families. This does not exclude the existence of species or families within the Nodosariata that may have a different

elemental signature than those reported here as we investigated 3 species only. Still, the analyzed species span 3 different

orders within the Nodosariata. The relative uniformity in shell carbonate composition across the orders may indicate that the

calcification mechanism invented by the Nodosariata is very well suited for a wide range of seawater chemical conditions.

Alternatively, the relatively low species diversity of the present day Nodosariata compared to that during the Jurassic

(Haynes 1981a) may reflect a selective loss of calcification mechanisms due to past changes in ocean chemistry and/or

physics, possibly related to past climate variability. Such a hypotheses on the potential interplay between calcification and

climate can be tested by comparing the El/Ca of extinct species from a suite of eras to that of species living today. (Haynes

1981b; 1981a)·(Evans et al. 2013; Maeda et al. 2017)·(Toyofuku et al. 2011; Barrientos et al. 2018).

### 4.2  Effect of environment in the element incorporation

#### 4.2.1    Na/Ca versus salinity

Na/Ca correlates with salinity in 2 of the Nodosariata species investigated: *Dentalina* spp. and *Nodosaria flintii* (Fig. 4).

Sensitivities of Na/Ca to salinity relationships appear somewhat higher than those reported for Rotaliida species (Wit et al.

2013; E Geerken et al. 2019; Allen et al. 2016; Mezger et al. 2016; Hauzer et al. 2021) (Table 3). Parallel to the increasing

number of reports on the correlation between Na incorporation and salinity, there is discussion of what precisely controls

foraminiferal Na/Ca, which could be (Hauzer et al. 2018) for seawater $[Ca^{2+}]_{sw}$ as well as the $[Na^{+}]_{sw}$ and/or salinity (Wit et

al. 2013). In addition, it may be that Na incorporation is affected by precipitation rates as well, such as indicated by

inorganic experiments showing that Na-incorporation is affected by saturation state (Devriendt et al. 2021). Although

poorly constrained in foraminifera (Geerken et al. 2022), environmental factors may affect the rate at which foraminifera

precipitate their calcite, making the relationship between Na/Ca and an environmental parameter indirect. Still, the

consistent increase in Na/Ca with salinity in many Rotaliida foraminifera and the here reported correlations for two



Nodosariata species (Fig 4; Table 3), suggests a more direct coupling between seawater [Na$^+$] and or [Ca$^{2+}$] and underscores the robustness of this proxy.

| Order | Species | Sensitivity (mmol/mol) | Paper |
|---|---|---|---|
| Nodosariida | *Nodosaria flintii* | 1.27 | This study |
| Nodosariida | *Dentalina* sp | 1.19 | This study |
| Rotaliida low Mg | *Ammonia tepida* | 0.22 | Wit et al., 2013 |
| Rotaliida low Mg | *Ammonia tepida* | 0.064 | Geerken et al., 2019 |
| Planktonic Rotaliida | *G. ruber* | 0.074 | Allen et al., 2016 |
| Rotaliida med Mg | *Amphistegina lessonii* | 0.077 | Geerken et al., 2019 |
| Planktonic Rotaliida | *G. ruber* | 0.66 | Mezguer et al., 2016 |
| Planktonic Rotaliida | *G. sacculifer* | 0.6 | Mezguer et al., 2016 |
| Rotaliida High Mg | *Operculina ammonoides* | 0.33 | Hauzer et al., 2021 |

**Table 3: Comparison of sensitivities of Na/Ca versus salinity of benthic foraminifera from this study and Rotaliida with different Mg incorporation ratios.**

### 4.2.2 Mg/Ca versus temperature

The Mg/Ca-temperature relationships found for the Nodosariata species reported here (Fig. 3) is likely also affected by different bottom water [$CO_3^{2-}$] at the sampled stations (Sadekov et al. 2014). The effect of saturation state on Mg-

incorporation was found to be approximately 40 mmol/mol for a change of ~1000 µmol [$CO_3^{2-}$]/kg seawater in culturing experiments (Dissard et al. 2010; Yu et al. 2019; Inge van Dijk et al. 2017). This would amount to a change of approximately 2 mmol/mol Mg/Ca over the total change at the three locations studied here, assuming that the sensitivity of Mg-

incorporation as a function of $[CO_3^{2-}]$ in the Nodosariata is similar to that in Rotaliida. This would reduce the change in Mg/Ca as a function of temperature by less than 10% and hence this would only have a very modest impact on the here

reported Mg/Ca-temperature sensitivities (Fig. 3).

For each of the three species that were found at all three stations, Mg/Ca increased exponentially with temperature (Fig 3). These Mg/Ca-temperature sensitivities are slightly lower than those reported for most Rotaliida species. On average, Mg/Ca increases by 6% per °C in the Nodosariata species analyzed here (Fig 3), while in many planktonic species Mg/Ca increases by 10% per °C (Barker et al. 2005). Low-Mg/Ca benthic rotaliids display an increase of approximately 8% in Mg/Ca

per °C (Lear, Rosenthal, and Slowey 2002; M. Raitzsch et al. 2010; Russell et al. 2004). High-Mg/Ca benthic Rotaliida species, on the other hand increases by only 2% (Maeda et al. 2017), which is similar to the slopes of those reported for Miliolida (L. J. de Nooijer et al. 2017; Toyofuku et al. 2000b).

Combining sensitivities for the different groups of foraminifera and their average Mg/Ca and comparing them to those of inorganically precipitated calcites (Morse, Arvidson, and Lüttge 2007; Wit et al. 2012), suggests a negative relation between

Mg-incorporation and sensitivity to temperature (Fig. 5). The relative increase in Mg/Ca with temperature is smaller for species incorporating relatively much Mg in their calcite and vice versa. Highest Mg/Ca ratios are found in inorganically precipitated calcites (Morse, Arvidson, and Lüttge 2007) in which the increase of Mg/Ca is approximately 2 % for a 1 °C temperature increase. For species incorporating equally much Mg (e.g. *Operculina ammonoides* (Evans et al. 2013)*)*, the slope of the Mg/Ca-temperature calibration is similar, while for species like *Ammonia tepida* (incorporating 50-100 times

less Mg in their shell), the increase is approximately 7% per 1 °C temperature increase (Fig. 5).

This suggests that the observed high sensitivity of Mg/Ca to temperature in the low Mg/Ca species actually consists of two factors: an inorganic temperature dependent fractionation and a biomineralization-related partitioning, which is also temperature dependent. The large difference in Mg/Ca between foraminifera (Wit et al. 2012) has been suggested to reflect the efficiency to lower the Mg/Ca in the calcifying fluid, either achieved by active $Mg^{2+}$-removal (Elderfield, Bertram, and

Erez 1996; Spero et al. 2015) or by selective inward $Ca^{2+}$ transport (Toyofuku et al. 2017). Foraminiferal species with calcite Mg/Ca ratios close to those found in inorganic precipitation experiments may well lack such a mechanism and the increase





in Mg/Ca with temperature hence matches that found in inorganic precipitation experiments. The species that are capable of

lowering the Mg/Ca in the fluid from which they calcify, incorporate consistently more Mg at increased temperatures (Fig.

5). This suggests that foraminiferal Mg/Ca-temperature relationships are determined by two components (Dämmer et al.

2021). The first component is the biological control on Mg partitioning and the second component is the thermodynamic

effect of temperature on Mg/Ca. In foraminiferal species where the first component is absent (i.e. when they precipitate

from a seawater like fluid), the second component determines the Mg/Ca-temperature sensitivity. For species that lower

the $Mg^{2+}$ in the fluid from which they precipitate their calcite, the biological component dominates the Mg/Ca-temperature

calibration. The relatively large variability in the low Mg/Ca species may be explained by small environmental factors (e.g.

salinity or water depth) or by processes that are part of the calcification mechanism (e.g. Rayleigh fractionation, organic

templates) that may vary slightly between species. The Nodosariata studied here have similar Mg/Ca, but differ in their

sensitivities (Fig. 5), which may well reflect the environmental and/ or calcification-related differences between species.

(Elderfield, Bertram, and Erez 1996; Branson et al. 2018).

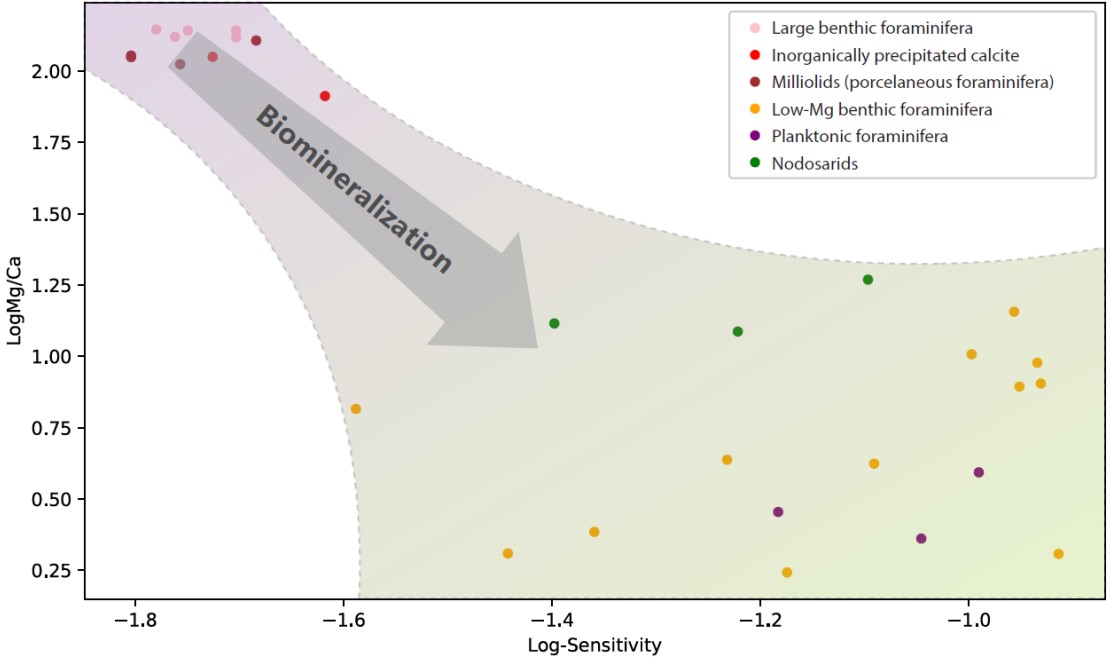
**Figure 5: Comparison between the sensitivity and the average Mg/Ca for different groups of foraminifera. Results for the Nodosariata are from this study, all other Mg/Ca-temperature calibrations are from previous studies (Toyofuku et al. 2011; Douglas and Staines-Urias 2007; Barrientos et al. 2018; Rosenthal, Boyle, and Slowey 1997; Quillmann et al. 2012; Markus Raitzsch et al. 2008; David W. Lea, Mashiotta, and Spero 1999; Anand, Elderfield, and Conte 2003; Barker et al. 2005; Rosenthal et al. 2011; Kristjánsdóttir et al. 2007; Lear, Rosenthal, and Slowey 2002; Evans et al. 2013; Maeda et al. 2017; Wit et al. 2012;**
**Morse, Arvidson, and Lüttge 2007; Toyofuku et al. 2000b; L. J. de Nooijer et al. 2017; Knorr et al. 2015). Species towards the lower right corner are increasingly affected by biomineralization. These calibrations were used to calculate and plot the Mg/Ca at 20 °C.**

## 5. CONCLUSION

The chemical composition of the shells of various Nodosariata species collected in the Gulf of Mexico was found to be
clearly different from those of other foraminiferal orders. Their Mg/Ca was between 6 and 20 mmol/mol and their Na/Ca was relatively high compared to ratios for most planktonic Rotaliida species. Sr/Ca and B/Ca were comparable to those found in other foraminiferal species. In two of the species studies, the Na/Ca increased linearly with salinity. Between families of the Nodosariata analyzed, the El/Ca was relatively similar. The Nodosariata's Mg/Ca is correlated to temperature and could thus serve as a sea water temperature proxy. Compared to Rotaliida and Miliolida orders our analysis show a
relation between the species' Mg/Ca and its sensitivity to changes in temperature. When more Mg is incorporated, it is less sensitive to changes in temperature and vice versa. This suggest the interaction between two components that together determine the Mg/Ca: the capacity of a species to control the Mg/Ca of the calcite and the thermodynamic effect of temperature on Mg incorporation.

## DATA AVAILABILITY

All raw data can be provided by the corresponding authors upon request (laura.pacho.sampedro@nioz.nl).

## AUTHOR CONTRIBUTIONS

Conceptualization, Gert-Jan Reichart and Lennart de Nooijer; generated geochemical proxy analyses, Laura Pacho; Interpretation of the data, Laura Pacho, with the support of the co-authors; Writing, Laura Pacho with the support of the co-authors. Gert-Jan Reichart secure funding for the project.

**COMPETING INTERESTS**

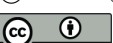
The contact author has declared that none of the authors has any competing interests.

**ACKNOWLEDGEMENTS**

We are grateful to the captain and crew on board of RV Pelagia expedition 64PE467 to the Gulf of Mexico. To Frans Jorissen for providing help with the identification of foraminifera. To Wim Boer for the LA-ICPMS analyses and Karel

Baker for the DIC and TA analyses.

**FINANCIAL SUPPORT**

This work was carried out under the program of the Netherlands Earth System Science Centre (NESSC), financially supported by the Ministry of Education, Culture and Science (OCW).

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
