# Peer review of "Element/Ca ratios in Nodosariida (Foraminifera) and their potential application for paleoenvironmental reconstructions"

_Biogeosciences, 2023_

## Referee Comment (RC1)

Element/Ca ratios in Nodosariida (Foraminifera) and their potential application for paleoenvironmental reconstructions

Review

In my opinion this is an overall excellent paper which should be published, with appeal to a broad audience, i.e.,. ranging paleoceanographers to geochemists working on carbonate-based proxy development to foraminifera-specialists. The generated data on the element/Ca values of 5 common used proxies (including calibration of the Mg/Ca temperature proxy) are of importance for reconstruction of paleoenvironments, to the opportunities and pitfalls of carbonate produced by living organisms, to undertsanding the phylogeny, evolution and relations between different groups of foraminifera.  The paper is of specific interest for reconstruction of late Paleozoic-early Mesozoic environment, because the studied group (Nodosariata) was a dominant group of benthic foraminifera before the Albian-Santonian radiation of the Rotaliida, presently by far the dominant, most highly diverse group of calcifying benthic foraminifera and all planktic foraminifera.

Various authors have included Nodasariata in their stable oxygen and carbon isotope analysis, though usually not defined at the species level. Such data commonly showed large errors, and/or did not correlate well with analysis of Rotaliida in the same samples. Could the authors say something about stable isotope data available for Nodosariata?

In view of the fact that such analyses usually combined different morphospecies of e.g. *Lenticulina*, could the authors specifically comment on differences in El/Ca by species within the same genus?

Methods:
Others might prefer somewhat different methodology (e.g., rose bengal stainin g, absence of reductive cleaning, check for clay contamination), but in my opinion the methodology is acceptable since clearly described, so readers can take into account how measurements were made. Statistical data analsyis is also well described and looks good.

Taxonomy:
Nodosariata taxonomy is complex, and definitely not full examined; in general the names cited with images in fig. 2 seem OK, except for 2d: *Lenticulina papillosa*, which species is a densely coiled spiral, not unrolling. A species with a uniserial part is *Vaginulinopsis baggi* McLean 1955 (https://www.marinespecies.org/aphia.php?p=taxdetails&id=896255).
In 'fundamentally distinct morphology' (line 197), I would like to see the very typical apertural morphology mentioned in addition to the chamber arrangement, because the aperture is easily recognizable for non-foram specialists.
line 203: 'large difference *in time* of first fossil occurrence' - please add 'in time'
line 206:' *'El/Ca* variability ...is remarkably small'; please add 'El/Ca' or 'chemistry of shell carbonmate', to make this clear (in fact, morphological variability is rather large)
line 215: very good to see that added.
lines 238-240: the studied station are all relatively shallow, so I would not expect significant undersaturation - please provide values of carbonate saturation so the reader can see what these values are.
lines 270-275: somewhat speculative, but I like this hypothesis of dual components.
Figure 5: please use different signature for inorganically precipitated calcite, to make that stand out.

I strongly suggest that the data are placed at an online data provides such as pangaea.de, rather than 'made available'.

---

## Referee Comment (RC2)

**Review of "Element/Ca ratios in Nodosariida (Foraminifera) and their potential application for paleoenvironmental reconstructions"**

Submitted by Pacho et al. to *Biogeosciences*

6/27/23

**Summary**: Pacho et al. submitted a very good manuscript that matches well the scope of *Biogeosciences*; it is suitable for publication after minor revisions.

Foraminifera are arguably one of the most important archives of the Earth's past climate, however, the incorporation of environmental signals into the shell chemistry is only well-studied for foraminifera of the order Rotaliida. Using other foraminifer orders for paleoclimate reconstructions such as the order of the Nodosariida would open the door to countless new applications. For example, Nodosariata appear much earlier in the fossil record than the Rotaliida, which could extend the range of foraminifera-based reconstructions significantly.

However, published data and proxy calibrations (in particular for trace element based proxies) for Nodosariata are very sparse to non-existent, thus, the study presented by Pacho et al. presenting B/Ca, Na/Ca, Mg/Ca, Sr/Ca and Ba/Ca measurements from Nodosariata shells (by LA-ICP-MS) and relationships between Mg/Ca and temperature as well as Na/Ca and salinity is a highly appreciated first step to expand the Mg/Ca temperature proxy (and Na/Ca salinity proxy) to a new and vastly underutilized order of foraminifera.

I have a couple of minor comments and remarks that I would like to see addressed and/or included in the paper before I consider this manuscript ready for publication:

**Discussion:**

The fact that Nodosariata appear in the Carboniferous provides researchers with the opportunity to extend the range of foraminifera-based climate reconstructions significantly (the authors are also mentioning this in the abstract as a motivation for this study). However, how realistic would it be to find Carboniferous shells that are well-enough preserved for paleoclimate reconstructions? Any prior studies assessing the preservation of these shells? I think this needs several sentences of discussion.

Also, there are many studies discussing the evolution of seawater Mg/Ca over time - this is relevant for the accuracy of the Mg/Ca temperature proxy in 'deep-time'. I believe that the discussion-section of this manuscript would benefit by the addition of a few sentences addressing this topic.

**Method:**

LA-ICPMS analyses of foraminifera shells are not trivial, and many papers have been published, discussing different approaches of analysis and methodologies to process the data. I am not able to find many information within this manuscript. For example, was the entire LA-ICPMS profile integrated, and the average element/Ca ratios calculated? Was the intrashell variability monitored, while the laser milled through the chamber wall? There could be some additional information in this signal. Where there any criteria for cut-off? (e.g., laser penetrates the chamber wall – at what point to end integration of the data). What type of laser was used, what was the wavelength (193 nm?). In addition, I had to google the NFHS-2-NP standard. Please add some info. Did the authors perform a standard-sample-standard bracketing approach? I assume they did, but I wasn't able to find this in the description.

**Figure 1:** The legends in the temperature and salinity plots are not clear. What are the symbols labeled "Nodosariida?" I am guessing the actual water depths of the sediment samples from which the shells were recovered. What are then the symbols labeled "sample_location"?

According to the text, samples were collected at 105 m, 272 m, and 619 m water depth. Thus, we should have data from only 3 depths. Why are more depth intervals in the plot?

Also, why are >20 red dots/markers in the map? This is not clear. According to table 1, there are just 3 stations where foraminifera, chemical and physical seawater parameters were collected. Then why so many dots? Please revise the legend and caption of Fig. 1, otherwise, it is very confusing.

**Some minor items to address:**

**Figures 3, 4:** please write rose-Bengal observed (or living) instead of rB observed

Line 58: it shall read: "...***that*** can be optically…"

Line 67: ~ 190 Ma

Line 73: suggest

Line 75: along ***a*** depth transect

Line 117: It sounds as if the repeatability is based on data from a different analytical session (Boer et al., 2022) using NFHS-2-NP standard. However, this study used the MACS-3 standard for calibration. What is the repeatability on MACS-3?

Line 117: It seems that 2-3 analysis were placed in the final chamber (Fig. 2). Please add this information here.

Line 135: **the** package

Line 136: Dot before "Identified" missing.

Line 157: over  the

Line 159: please also mention the water temperatures, as temperatures are more relevant than the water depth (correlation between Mg/Ca and temperature, not water depth)

Line 206: variability **in the chemical composition**

Lines 206 – 218: please add some discussion of past seawater Mg/Ca and the implication on 'deep-time' T-reconstructions

Line 225  seawater

Line 242 range instead of change?

In addition, I strongly recommend that the authors upload their data to a open-access database such as Pangaea.de.

---

## Author Response (AR3)

- We thank Dr. Ellen Thomas for her comments on our manuscript. Below, we will address all her comments and suggestions (in bold) one-by-one:
1. **Various authors have included Nodasariata in their stable oxygen and carbon isotope analysis, though usually not defined at the species level. Such data commonly showed large errors, and/or did not correlate well with analysis of Rotaliida in the same samples. Could the authors say something about stable isotope data available for Nodosariata?**

We didn't analyze carbon and oxygen isotopes but differences in shell chemistry between Rotaliida and Nodosariata can be explained by the differences in their biomineralization mechanisms. In a recent paper, Dubicka et al. (2021) show that the oxygen isotopic difference between the two groups are relatively small, despite the marked differences in El/Ca between the groups.

The small carbon isotope difference between the two types of foraminifera (Dubicka et al., 2021) could result from their carbon-uptake mechanism, but may also be caused by their ecology: when in-sediment depth habitats differ between species (i.e. epi- versus infauna), offsets in the isotopic composition can occur. For instance, a less negative $\delta^{13}C$ value may be observed in species calcifying closer to the sediment-water interface because less metabolic DIC is present there. Here we did not include the stable oxygen and carbon isotopes in our analytical protocol as we needed -after the laser ablation analyses reported here- to dissolve the shells for another (ongoing) study. We have added this reference and a short description of the known similarity in $\delta^{18}O$ and small difference in $\delta^{13}C$ to the Introduction (lines 72-75).

2. **In view of the fact that such analyses usually combined different morphospecies of e.g. *Lenticulina*, could the authors specifically comment on differences in El/Ca by species within the same genus?**
For the two species within the genus of *Lenticulina* the data show a significant difference in Mg/Ca at 100 m between *L. calcar* and *L. denticulifera* using a t-test assuming different variances. At a depth of 300m there were no significant differences between these two species in either Mg/Ca or Na/Ca. We now added the results of the statistical analysis to the supplementary material, showing that in our samples the El/Ca between species of *Lenticulina* at the highest bottom-water temperature was significantly different, while at the deeper station, Mg/Ca did not vary significantly between species. Overall, differences in Mg/Ca between species, or variability in the total Nodosariid population, is relatively small (3 – 30 mmol/mol).

| Depth (m) | *L. calcar* vs *L. denticulifera* (t-test 2 tailed) Mg/Ca (mmol/mol) | *L. calcar* vs *L. denticulifera* (t-test 2 tailed) Na/Ca (mmol/mol) |
|---|---|---|
| 104.7 | p-value < 0.05 | p-value > 0.05 |
| 271.6 | p-value > 0.05 | p-value < 0.05 |

We also added a table showing the differences within species at the places where we found sufficient rB-stained samples. A t-test showed that the variability in *Dentalina* spp. is significant only at 271 m, for the other 2 depths, the differences in Mg/Ca are not significant.

| Depth (m) Mg/Ca | *Dentalina* spp. t-test | *Lenticulina calcar* t-test | *Nodosaria flintii* t-test |
|---|---|---|---|
| 104.7 | p-value > 0.05 | All living | p-value > 0.05 |
| 271.64 | p-value < 0.05 | All living | Non living |
| 618.8 | p-value > 0.05 | Non living | Non living |

| Depth (m) Na/Ca | *Dentalina* spp. t-test | *Lenticulina calcar* t-test | *Nodosaria flintii* t-test |
|---|---|---|---|
| 104.7 | p-value > 0.05 | All living | p-value > 0.05 |
| 271.64 | p-value < 0.05 | All living | Non living |
| 618.8 | p-value < 0.05 | Non living | Non living |

**Methods:**
3. **Others might prefer somewhat different methodology (e.g., rose bengal staining, absence of reductive cleaning, check for clay contamination), but in my opinion the methodology is acceptable**

**since clearly described, so readers can take into account how measurements were made. Statistical data analysis is also well described and looks good.**

We did use rose Bengal staining, but that did not yield sufficient stained specimens for our purposes. We compared the stained to the unstained specimens, which showed that there is no systematic offset in elemental composition between the two groups (see table above). We checked the possible necessity of a reductive cleaning by analysis of individual laser ablation profiles, but did not find evidence for higher (trace) element concentrations at the outside surface of the shell. Subsequently, for the data processing, a general quality check of the data was performed by correlating Mg/Al and Mg/Mn, not only in the whole data but between groups and species, showing that the Mg/Ca data was not impacted by Mn-oxides. This information is added to the data that that will be made available (also following the last comment in this review).

**4. Taxonomy:**
**Nodosariata taxonomy is complex, and definitely not full examined; in general, the names cited with images in fig. 2 seem OK, except for 2d: *Lenticulina papillosa*, which species is a densely coiled spiral, not unrolling. A species with a uniseral part is *Vaginulinopsis baggi* McLean 1955. (hZps://www.marinespecies.org/aphia.php?p=taxdetails&id=896255).**

We thank the reviewer for this suggestion and have changed this throughout the manuscript.

**5. In 'fundamentally distinct morphology' (line 197), I would like to see the very typical apertural morphology mentioned in addition to the chamber arrangement, because the aperture is easily recognizable for non-foram specialists.**

We tried to orientate the shells as to optimize the view for both chamber arrangement and showing the aperture. A more detailed picture of the aperture for most species is included in the new revised manuscript. (Added blow up of the aperture in the supplementary material).

**6. line 203: 'large difference *in time* of first fossil occurrence' - please add 'in time'**
This has been added (now line 210).

**7. line 206:' *El/Ca* variability ...is remarkably small'; please add 'El/Ca' or 'chemistry of shell carbonate', to make this clear (in fact, morphological variability is rather large)**
We agree and have added this to the text (now line 215)

**8. line 215: very good to see that added.**

**9. lines 238-240: the studied station are all relatively shallow, so I would not expect significant undersaturation - please provide values of carbonate saturation so the reader can see what these values are.**

The omega values (i.e. degree of super-saturation) were already in the original manuscript (Table 1). Even at the deepest station, bottom waters were supersaturated with respect to calcite (1.75).

**10. lines 270-275: somewhat speculative, but I like this hypothesis of dual components.**

**11. Figure 5: please use different signature for inorganically precipitated calcite, to make that stand out.**
We now used a diamond for the inorganically calibrated Mg/Ca-temperature relationship.

- We would like to thank Dr Reinhard Kozdon for his constructive comments, which we will incorporate into the next version of our manuscript. Below we copied his original comments (in bold) and added our answers one-by-one.

**Foraminifera are arguably one of the most important archives of the Earth's past climate, however, the incorporation of environmental signals into the shell chemistry is only well-studied for foraminifera of the order Rotaliida. Using other foraminifer orders for paleoclimate reconstructions such as the order of the Nodosariid a would open the door to countless new applications. For example, Nodosariata appear much earlier in the fossil record than the Rotaliida, which could extend the range of foraminifera-based reconstructions significantly.**

**However, published data and proxy calibrations (in particular for trace element based proxies) for Nodosariata are very sparse to non-existent, thus, the study presented by Pacho et al. presenting B/Ca, Na/Ca, Mg/Ca, Sr/Ca and Ba/Ca measurements from Nodosariata shells (by LA-ICP-MS) and relationships between Mg/Ca and temperature as well as Na/Ca and salinity is a highly appreciated first step to expand the Mg/Ca temperature proxy (and Na/Ca salinity proxy) to a new and vastly underutilized order of foraminifera.**

**I have a couple of minor comments and remarks that I would like to see addressed and/or included in the paper before I consider this manuscript ready for publication:**

**Discussion:**

1. **The fact that Nodosariata appear in the Carboniferous provides researchers with the opportunity to extend the range of foraminifera-based climate reconstructions significantly (the authors are also mentioning this in the abstract as a motivation for this study). However, how realistic would it be to find Carboniferous shells that are well-enough preserved for paleoclimate reconstructions? Any prior studies assessing the preservation of these shells? I think this needs several sentences of discussion.**

We agree with the reviewer that finding well preserved shells indeed becomes increasingly challenging further back in geological time. We added to the text (lines 254-255): "Using the chemical composition of fossil shells this far back in time requires careful assessment of the calcite for diagenetic overprints."

2. **Also, there are many studies discussing the evolution of seawater Mg/Ca over time - this is relevant for the accuracy of the Mg/Ca temperature proxy in 'deep-time'. I believe that the discussion-section of this manuscript would benefit by the addition of a few sentences addressing this topic.**

We agree that the current uncertainty in reconstructed seawater Mg/Ca and the combined influences of temperature and seawater Mg/Ca on foraminiferal Mg/Ca (particularly in extinct species) hampers accurate and precise temperature reconstructions. However, seawater $[Ca^{2+}]$ and $[Mg^{2+}]$ reconstructions may well improve in the future and so will the mechanistic understanding of Mg-incorporation improve. We decided to keep the potential of T-reconstructions in deep time, but have added some cautions (lines 256-258) when aiming for such reconstructions.

**Method:**

3. **LA-ICPMS analyses of foraminifera shells are not trivial, and many papers have been published, discussing different approaches of analysis and methodologies to process the data. I am not able to find many information within this manuscript. For example, was the entire LA-ICPMS profile integrated, and the average element/Ca ratios calculated? Was the intrashell variability monitored, while the laser milled through the chamber wall? There could be some additional information in this signal. Where there any criteria for cut-off? (e.g., laser penetrates the chamber wall – at what point to end integration of the data).**

We did explain this somewhat in the manuscript already (lines 113 to 124) and we refer to earlier publications on this subject from our group. To accommodate the reviewers´ comments we now added: For average calculating average chamber elemental ratios we integrated the entire laser ablation profile, which was monitored for each measurement.

| Laser ablation system sample cell | NIOZ |
|---|---|
| Wavelength | 193nm |
| Pulse duration | 4 ns |

| | |
|---|---|
| Laser fluence | 1 J/cm$^2$ |
| Laser spot size | 60 µm |
| Laser repetition rate | 6 Hz |
| Carrier gas flow rate (He) | 0.6 L/min |

4. **What type of laser was used, what was the wavelength (193 nm?). In addition, I had to google the NFHS-2-NP standard. Please add some info. Did the authors perform a standard-sample-standard bracketing approach? I assume they did, but I wasn't able to find this in the description.**

For the NFHS-2-NP standard we referred to Boer et al. (2022). This publication lists the methods used and data of all laboratory involved in obtaining the reference data used for calibration. This paper also gives the settings and equipment used in the analyses, including the exact settings and standard bracketing and statistical data treatment. For completeness and easy access we here now added a short overview of the most important parameters. "The wavelength of the LA-ICPMS used is deep ultra violet (193 nm), which is excellently suited for ablating carbonates (Reichart et al., 2003).

5. **Figure 1: The legends in the temperature and salinity plots are not clear. What are the symbols labelled "Nodosariida?" I am guessing the actual water depths of the sediment samples from which the shells were recovered. What are then the symbols labeled "sample_location"?**

This figure was not very clear and we have now included an improved version of this figure. We removed the other stations we investigated and we did not find enough nodosarids. Also, we have indicated the location of the CTD station used for the temperature and salinity profile. In the depth profile we have only added those station which were used for our calibration.

6. **According to the text, samples were collected at 105 m, 272 m, and 619 m water depth. Thus, we should have data from only 3 depths. Why are more depth intervals in the plot?**

We have changed this accordingly, see also answer to previous comment.

7. **Also, why are >20 red dots/markers in the map? This is not clear. According to table 1, there are just 3 stations where foraminifera, chemical and physical seawater parameters were collected. Then why so many dots? Please revise the legend and caption of Fig. 1, otherwise, it is very confusing.**

We have changed this based on the comment. We agree that this was not very clear. See also to previous comments/answers.

**Some minor items to address:**

8. **Figures 3, 4: please write rose-Bengal observed (or living) instead of rB observed**

We will change this accordingly.

9. **Line 58: it shall read: "...that can be optically…"**

Correct: now changed 'than' to ' that'.

10. **Line 67: ~ 190 Ma**

Agreed and changed accordingly.

11. **Line 73: suggest**

Changed

12. **Line 75: along a depth transect**

Added 'a'.

13. **Line 117: It sounds as if the repeatability is based on data from a different analytical session (Boer et al., 2022) using NFHS-2-NP standard. However, this study used the MACS-3 standard for calibration. What is the repeatability on MACS-3?**

*Repeatability based on RSDs of measurements of NFHS-2-NP in this study (n=8) is 4% for Na/Ca, 1% for Mg/Ca and 2% for Sr/Ca. We have also added the variability in MAC-3 in the revised version of our manuscript. ¨Calibration was performed against USGS MACS-3 (synthetic calcium carbonate) pressed powder standard with $^{43}Ca$ as an internal standard.¨*

**14. Line 117: It seems that 2-3 analysis were placed in the final chamber (Fig. 2). Please add this information here.**

We have added this to the new paragraph that describes the analytical procedure in more detail (124) and Fig. 2.

**15. Line 135: the package**

Added.

**16. Line 136: Dot before "Identified" missing.**

Inserted.

**17. Line 157: over a the**

'the was removed' (now line 175).

**18. Line 159: please also mention the water temperatures, as temperatures are more relevant than the water depth (correlation between Mg/Ca and temperature, not water depth).**

*Bottom water temperatures are now added to Table 1.*

**19. Line 206: variability in the chemical composition**

This was also noted by the other reviewer: it now reads: 'variability in the element-to-calcium ratio'

**20. Lines 206 – 218: please add some discussion of past seawater Mg/Ca and the implication on 'deep-time'**

*We have added a small section on changes in sea water chemistry on geological time scales and potential impacts on Mg/Ca- T reconstructions. This section now starts with the added cautionary note on preservation of the original signal in Nodosarids (or in foraminiferal calcite in general) from consecutively old ages.*

**21. Line 225 for seawater**

Added.

**22. Line 242 range instead of change?**

Added.

**23. In addition, I strongly recommend that the authors upload their data to a open-access database such as Pangaea.de.**

*Royal NIOZ has a strict data access policy following the FAIR principles. The NIOZ has its own repository and a DOI will be providing full access to the data published.*

**List of changes:**

| Dr. Ellen Thomas question number | Manuscript_Lines | Comments |
|---|---|---|
| 1 | 72-75 | |
| 2 | 159 | Table S2 |
| 3 | 169 | Variability between living and non-living |
| | 129 | Table S1 |
| 4 | 136 | And Table 3 |
| 5 | 140 | and Fig. S5 |
| 6 | 210 | Changed |
| 7 | 215 | Changed |
| 9 | Table 1 | |
| 11 | Fig. 5 | |
| **Dr. Reinhard Kozdon** | **Manuscript_Lines** | **Comments** |
| 1 | 254-255 | |
| 2 | 256-258 | |
| 3 | Table 2 | |
| 4 | 119 | |
| 5, 6 and 7 | Fig. 1 | |
| 8 | Fig. 3, 4 | |
| 9 | 57-58 | |
| 10 | 65 | |
| 11 | 71 | |
| 12 | 75 | |
| 13 | 116-118 | |
| 14 | 124 | And Fig. 2 |
| 15 | 146 | |
| 16 | 147 | |
| 17 | Changed | |
| 18 | Table 1 | |
| 19 | 215 | |
| 20 | 256-258 | |
| 21 | Added | |
| 22 | 248 | |
| 23 | 310 | |
| **Editor Chiara Borrelli** | **Manuscript_Lines** | **Comments** |
| 1 | 79, 104 and 112 | |
| 2 | Table 3, 163-165 | |
| 3 | Fig. S5 | |
| **Editor Chiara Borrelli (2)** | **Manuscript_Lines** | **Comments** |
| 1 | 71-76 | |
| 2 | Fig. 1 | |
| 3 | Table 1 | Figure capture |
| 4 | 121-122 | |
| 5 | 99 | Table 1 |
| 5 | 119 | Table 2 |

| | Manuscript_Lines | Comments |
|---|---|---|
| 6 | 149-150 | |
| 7 | 220 | Figure in the supplementary material |
| 8 | 266 | |
| 9 | 268 | |
| 10 | 201-205; 211-213 | |
| **Editor Chiara Borrelli (3)** | **Manuscript_Lines** | **Comments** |
| 1 | 128-129 | |
| 2 | 166 | |
| 3 | 170 | |